# GNN-RAG: Graph Neural Retrieval for Large Language Model Reasoning

## Abstract

Retrieval-augmented generation (RAG) in Knowledge Graph Question Answering (KGQA) enriches the context of Large Language Models (LLMs) with retrieved KG information based on the question. However, KGs contain complex graph information and existing KG retrieval methods are challenged when questions require multi-hop information. To improve RAG in complex KGQA, we introduce the GNN-RAG framework, which leverages Graph Neural Networks (GNNs) for effective graph reasoning and retrieval. GNN-RAG consists of a *graph neural* phase, where the GNN retriever *learns* to identify useful graph information for KGQA, e.g., when tackling complex questions. At inference time, the GNN scores answer candidates for the given question and the shortest paths in the KG that connect question entities and answer candidates are retrieved to represent KG reasoning paths. The paths are verbalized and given as context to the downstream LLM for ultimate KGQA; GNN-RAG can be seamlessly integrated with different LLMs for RAG. Experimental results show that GNN-RAG achieves state-of-the-art performance in two widely used KGQA benchmarks (WebQSP and CWQ), outperforming or matching GPT-4 performance with a 7B tuned LLM. In addition, GNN-RAG excels on multi-hop and multi-entity questions outperforming competing approaches by 8.9–15.5% points at answer F1. Furthermore, we show the effectiveness of GNN-RAG in retrieval augmentation, which further boosts KGQA performance.

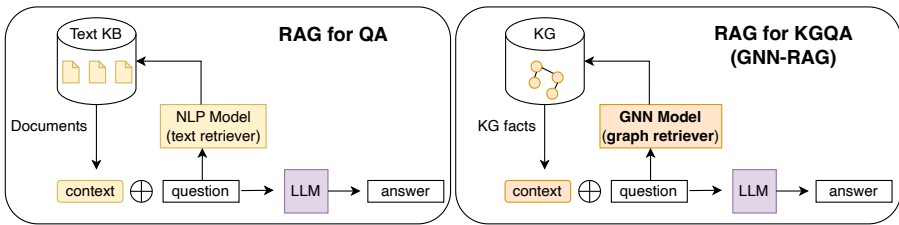

Figure 1: GNN-RAG leverages GNNs for retrieval over KGs (right), similar to how conventional text-based RAG works (left).

## 1 Introduction

Large Language Models (LLMs) (Brown et al., 2020; Bommasani et al., 2021; Chowdhery et al., 2023) are the state-of-the-art models in many NLP tasks due to their remarkable ability to understand natural language. LLM power stems from pretraining on large corpora of textual data to obtain general human knowledge (Kaplan et al., 2020; Hoffmann et al., 2022). However, because pretraining is costly and time-consuming (Gururangan et al., 2020), LLMs cannot easily adapt to new or in-domain knowledge and are prone to hallucinations (Zhang et al., 2023b).

Knowledge Graphs (KGs) (Vrandečić & Krötzsch, 2014) are databases that store information in structured form that can be easily updated. KGs represent human-crafted factual knowledge in the form of triplets *(head, relation, tail)*, e.g., `<Jamaica → language_spoken → English>`, which collectively form a graph. In the case of KGs, the stored knowledge is updated by fact addition or removal. As KGs capture complex interactions between the stored entities, e.g., multi-hop relations,

they are widely used for knowledge-intensive task, such as Question Answering (QA) (Pan et al., 2024).

Retrieval-augmented generation (RAG) is a framework that alleviates LLM hallucinations by enriching the input context with up-to-date and accurate context (Lewis et al., 2020), e.g., documents retrieved by a text knowledge base (KB) or facts retrieved from a KG; see Figure 1. In the KGQA task, the goal is to answer natural questions grounding the reasoning to the information provided by the KG. For instance, the context for RAG becomes "`Knowledge: Jamaica → language_spoken → English \n Question: Which language do Jamaican people speak?`", where the LLM has access to KG information for answering the question.

RAG's performance highly depends on the KG facts that are retrieved (Wu et al., 2023) and the challenge in KGQA is that KGs store complex graph information (they usually consist of millions of facts). Retrieving the right information requires effective graph understanding, while retrieving irrelevant information may confuse the LLM during KGQA reasoning (Wu et al., 2023). Existing retrieval methods that rely on off-the-shelf NLP retrievers (Baek et al., 2023) or classical graph algorithms (He et al., 2024) are limited as retrieval is not tailored for KGQA. On the other hand, graph retrieval powered by LLMs, such translating the question to relation paths (Luo et al., 2024) and traversing the KG guided by LLMs (Sun et al., 2024), is more effective but with certain challenges. Question translation depends on the LLM generating executable graph queries, while LLM-guided KG traversal requires a large number of LLM calls, which is limiting in production cases.

To address the limitations in retrieval for KGQA, we introduce GNN-RAG, a graph neural retrieval framework which is optimized for KGQA and can be seamlessly integrated with different downstream LLMs, similar to how conventional text-based RAG works (Figure 1). GNN-RAG relies on Graph Neural Networks (GNNs) (Mavromatis & Karypis, 2022), which are powerful graph representation learners, to handle the complex graph information stored in the KG for retrieval. GNN-RAG consists of a *graph neural* phase, where the GNN *learns* to identify useful graph information for KGQA, e.g., when tackling complex questions. At inference time, the GNN scores answer candidates for the given question and the shortest paths in the KG that connect question entities and answer candidates are retrieved, which are verbalized and given as context to

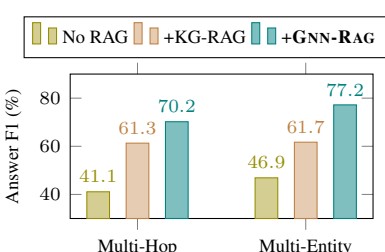

Figure 2: Retrieval effect on multi-hop/entity KGQA. Our **GNN-RAG outperforms** existing KG-RAG methods by 8.9–15.5% points at F1.

the LLM. Experimental results show GNN-RAG's superiority over competing RAG-based systems for KGQA by outperforming them by up to 15.5% points at complex KGQA performance (Figure 2). Furthermore, we show the effectiveness of GNN-RAG in retrieval augmentation, further boosting KGQA performance. Our **contributions** are summarized below:

- **Framework**: GNN-RAG leverages SOTA GNNs in KG retrieval to enhance RAG for KGQA. In our GNN-RAG framework, the GNN is optimized to retrieve useful graph information for KGQA, while the LLM leverages its natural language processing ability for ultimate KGQA. Similar to retrieval in text-based RAG, GNN-RAG can be seamlessly integrated with different downstream LLMs.

- **Effectiveness & Faithfulness**: GNN-RAG achieves state-of-the-art performance in two widely used KGQA benchmarks (WebQSP and CWQ). GNN-RAG retrieves multi-hop information that is necessary for faithful LLM reasoning on complex questions (8.9–15.5% improvement; see Figure 2). Moreover, GNN-RAG with retrieval augmentation further boosts KGQA performance.

- **Efficiency**: GNN-RAG improves vanilla LLMs on KGQA performance without incurring additional LLM calls as previous state-of-the-art RAG systems for KGQA require. In addition, GNN-RAG outperforms or matches GPT-4 performance with a 7B tuned LLM.

## 2 RELATED WORK

**KGQA Methods**. KGQA methods fall into two categories (Lan et al., 2022): (i) semantic parsing (SP) methods and (ii) information retrieval (IR) methods. SP methods (Sun et al., 2020; Lan & Jiang, 2020; Ye et al., 2022) learn to transform the given question into a query of logical form, e.g., SPARQL query. The transformed query is then executed over the KG to obtain the answers. However, SP methods require ground-truth logical queries for training, which are time-consuming to annotate in practice, and may lead non-executable queries due to syntactical or semantic errors (Das et al., 2021; Yu et al., 2022). IR methods (Sun et al., 2018; 2019; Zhang et al., 2022b) focus on the weakly-supervised KGQA setting, where only question-answer pairs are given for training. IR methods retrieve KG information, e.g., a KG subgraph (Zhang et al., 2022a), which is used as input during KGQA reasoning. GNN-RAG falls in the IR category.

**GNNs & LMs**. Combining GNNs with LMs has been the subject of a substantial body of existing literature (Jin et al., 2023), with various applications ranging from QA (Yasunaga et al., 2021; Wang et al., 2021; Zhang et al., 2022c; Tian et al., 2024; He et al., 2024; Zhang et al., 2024a) to training LMs on graphs (Zhao et al., 2022; Yasunaga et al., 2022; Huang et al., 2024). Such approaches seek to combine the natural language and graph reasoning into a single model by fusing *latent* GNN information with the LM. However, due to the modality mismatch of GNNs and LMs, fusing graph and natural language information is challenging for many knowledge-intensive tasks, even in supervised settings (Mavromatis et al., 2024). To alleviate this challenge, GNN-RAG divides KGQA in two stages. The GNN first retrieves useful information from the graph modality, which is then converted into natural language for effective LLM reasoning.

**GraphRAG**. GraphRAG usually refers to the general approach of inserting *verbalized* graph information at the context of LLMs (Peng et al., 2024; Wei et al., 2024) or leveraging additional graph information when retrieving context for RAG (Edge et al., 2024; Gutiérrez et al., 2024). For instance, verbalizing graph information obtained by KGs has been widely applied in GraphRAG (Xie et al., 2022; Baek et al., 2023; Jiang et al., 2023a; Jin et al., 2024; Liu et al., 2024). However, GraphRAG performance downgrades when the graph information retrieved is noisy and irrelevant to the question (Wu et al., 2023; He et al., 2024). To improve retrieval in KGQA, GNN-RAG employs a graph neural framework, which tailors graph retrieval for the KG at hand. By optimizing GNNs to identify the right graph information for answering the questions, GNN-RAG achieves superior retrieval performance compared to existing approaches in KGQA.

## 3 PROBLEM STATEMENT & BACKGROUND

**KGQA**. We are given a KG $\mathcal{G}$ that contains facts represented as $(v, r, v')$, where $v$ denotes the head entity, $v'$ denotes the tail entity, and $r$ is the corresponding relation between the two entities. Given $\mathcal{G}$ and a natural language question $q$, the task of KGQA is to extract a set of entities $\{a_q\} \in \mathcal{G}$ that correctly answer $q$. Following previous works (Lan et al., 2022), question-answer pairs are given for training, but not the ground-truth paths that lead to the answers.

**Retrieval & Reasoning**. As KGs usually contain millions of facts and nodes, a smaller question-specific subgraph $\mathcal{G}_q$ is retrieved for a question $q$, e.g., via entity linking and neighbor extraction (Yih et al., 2015). Ideally, all correct answers for the question are contained in the retrieved subgraph, $\{a_q\} \in \mathcal{G}_q$. The retrieved subgraph $\mathcal{G}_q$ along with the question $q$ are used as input to a reasoning model, which outputs the correct answer(s). The prevailing reasoning models for the KGQA setting studied are GNNs and LLMs.

**GNNs**. KGQA can be regarded as a node classification problem, where KG entities are classified as answers vs. non-answers for a given question. GNNs Kipf & Welling (2016); Veličković et al. (2017); Schlichtkrull et al. (2018) are powerful graph representation learners suited for tasks such as node classification. GNNs update the representation $\boldsymbol{h}_v^{(l)}$ of node $v$ at layer $l$ by aggregating messages $\boldsymbol{m}_{vv'}^{(l)}$ from each neighbor $v'$. During KGQA, the message passing is also conditioned to the given question $q$ (He et al., 2021). For readability purposes, we present the following GNN update for KGQA,

$$\boldsymbol{h}_v^{(l)} = \psi\Big(\boldsymbol{h}_v^{(l-1)}, \sum_{v' \in \mathcal{N}_v} \omega(q, r) \cdot \boldsymbol{m}_{vv'}^{(l)}\Big), \tag{1}$$

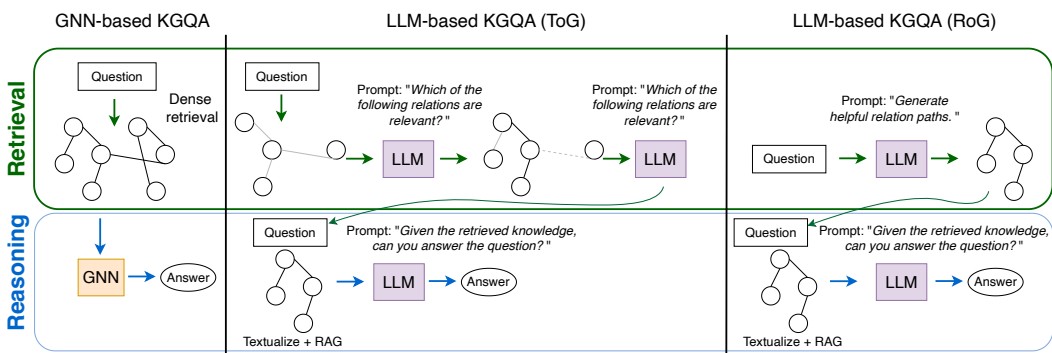

Figure 3: The landscape of existing KGQA methods. GNN-based methods reason on dense subgraphs as they can handle complex and multi-hop graph information. LLM-based methods employ the same LLM for both retrieval and reasoning due to its ability to understand natural language.

where function $\omega(\cdot)$ is typically a LM that measures how relevant relation $r$ of fact $(v, r, v')$ is to question $q$. Neighbor messages $\boldsymbol{m}_{vv'}^{(l)}$ are aggregated by a sum-operator $\sum$ and function $\psi(\cdot)$ combines representations from consecutive GNN layers.

**LLM RAG**. Retrieval-Augment Generation (RAG) is a method aiming to reduce LLM hallucinations (Lewis et al., 2020). Given a query $q$, RAG retrieves relevant information (e.g, documents from the given corpus), which is inserted as additional context $c$ to the LLM's input. In text applications, RAG leverages NLP models to identify relevant information (Karpukhin et al., 2020), such as retrieving the top-$k$ most semantic similar documents to the question, i.e, $c = [d_1, \ldots, d_k] = \text{top-}k_{d_i \in \mathcal{D}} M(d_i, q)$, where $\mathcal{D}$ is the document corpus and $M$ is the NLP model scoring.

In KGs, the context $c$ consists of graph information relevant to the question, such KG triplets, paths, or subgraphs. The retrieved graph information is first converted into natural language so that it can be processed by the LLM. The input given to the LLM contains the KG factual information along with the question and a prompt. For instance, the input becomes "`Knowledge: Jamaica → language_spoken → English \n Question: Which language do Jamaican people speak?`", where the LLM has access to KG information for answering the question.

**Landscape of KGQA methods**. Figure 3 presents the landscape of existing KGQA methods with respect to KG retrieval and reasoning. GNN-based methods, such as GraftNet (Sun et al., 2018), NSM (He et al., 2021), and ReaRev (Mavromatis & Karypis, 2022), reason over a dense KG subgraph leveraging the GNN's ability to handle complex graph information. Recent LLM-based methods leverage the LLM's power for both retrieval and reasoning (Gu et al., 2023). For instance, ToG (Sun et al., 2024) uses the LLM to retrieve relevant facts hop-by-hop. RoG (Luo et al., 2024) uses the LLM to generate plausible relation paths which are then queried on the KG to retrieve the relevant information.

**LLM-based Retriever**. We present an example of an LLM-based retriever (RoG; (Luo et al., 2024)). Given training question-answer pairs, RoG extracts the shortest paths to the answers starting from question entities for fine-tuning the retriever. Based on the extracted paths, an LLM (LLaMA2-Chat-7B (Touvron et al., 2023)) is fine-tuned to generate reasoning paths given a question $q$ as

$$\text{LLM}(\text{prompt}, q) \implies \{r_1 \to \cdots \to r_t\}_k, \tag{2}$$

where the prompt is "`Please generate a valid relation path that can be helpful for answering the following question: {Question}`". Beam-search decoding is used to generate $k$ diverse sets of reasoning paths for better answer coverage, e.g., relations {`<official_language>`, `<language_spoken>`} for the question "`Which language do Jamaican people speak?`". The generated paths are queried on the KG, starting from the question entities, in order to retrieve the intermediate entities for RAG, e.g., `<Jamaica → language_spoken → English>`.

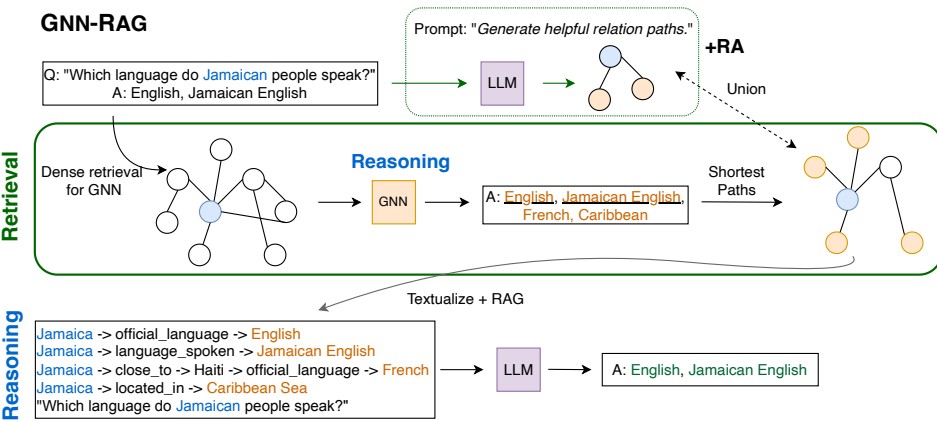

Figure 4: GNN-RAG: The GNN reasons over a dense subgraph to retrieve candidate answers, along with the corresponding reasoning paths (shortest paths from question entities to answers). The retrieved reasoning paths –optionally combined with retrieval augmentation (RA)– are verbalized and given to the LLM for RAG.

## 4 GNN-RAG

We introduce GNN-RAG, a novel graph neural retrieval method for KGQA that leverages state-of-the-art GNNs to improve retrieval performance when questions require complex graph information. We provide the overall framework at inference time in Figure 4. First, the KGQA GNN reasons over a dense KG subgraph to retrieve answer candidates for a given question. Second, the shortest paths in the KG that connect question entities and GNN-based answers are extracted to represent useful KG reasoning paths. The extracted paths are verbalized and given as context for LLM reasoning via RAG. In our GNN-RAG framework, the GNN acts as a dense subgraph reasoner to extract useful graph information, while the LLM leverages its natural language processing ability for ultimate KGQA.

### 4.1 GNN

In order to retrieve high-quality reasoning paths via GNN-RAG, we leverage state-of-the-art GNNs for KGQA. We prefer GNNs over other KGQA methods, e.g., embedding-based methods (Saxena et al., 2020), due to their ability to handle complex graph interactions and answer multi-hop questions. GNNs mark themselves as good candidates for retrieval due to their architectural benefit of exploring diverse reasoning paths (Choi et al., 2024) that result in high answer recall.

**GNN Optimization**. GNN reasoning consists of $L$ GNN updates via Equation 1 ($L$ is hyperparameter), where the node representations in the subgraph $\mathcal{G}_q$ are updated to $\boldsymbol{h}_v^{(L)}$. Given training question-answer pairs, the GNN is trained via node classification, where nodes have label $y_v = 1$ if they belong to the answer set $v \in \{a_q\}$ and $y_v = 0$, otherwise. The GNN parameters are optimized so that the nodes are scored as answers vs. non-answers based on their final GNN representations $\boldsymbol{h}_v^{(L)}$, followed by the $\mathrm{softmax}(\cdot)$ operation.

During inference, the nodes with the highest probability scores, e.g., above a probability threshold, are returned as candidate answers, along with the shortest paths connecting the question entities with the candidate answers (reasoning paths). The retrieved reasoning paths are used as input for LLM-based RAG.

**GNN Design**. Different GNNs may fetch different reasoning paths for RAG. To tackle multi-hop questions, we need an increased number of $L$ GNN layers, which we study in Section 4.4. As a result, we prefer deep GNNs, such as ReaRev (Mavromatis & Karypis, 2022), which allow to explore multi-hop paths to achieve high answer recall.

In addition, as presented in Equation 3, GNN reasoning depends on the question-relation matching operation $\omega(q, r)$. A common implementation of $\omega(q, r)$ is $\phi(\boldsymbol{q} \odot \boldsymbol{r})$ (He et al., 2021), where function $\phi$ is a neural network, and $\odot$ is the element-wise multiplication. We compute $K$ different question

representations $\boldsymbol{q}_k, k \in [0, K]$. Both questions and KG relations are encoded via a shared pretrained LM (Jiang et al., 2023b) as

$$\boldsymbol{q}_k = \gamma_k\big(\text{LM}(q)\big), \quad \boldsymbol{r} = \gamma_c\big(\text{LM}(r)\big), \tag{3}$$

where $\gamma_k$ is an attention-based pooling neural network that attends to question tokens, and $\gamma_c$ is the `[CLS]` token pooling. We provide the GNN implementation in Section 4.2.

In Appendix C, we develop a Theorem that shows that the GNN's output depends on the question-relation matching operation $\omega(q, r)$ and as result, we employ different LMs in Equation 3. Specifically, we train two separate GNN models, one using pretrained LMs, such as SBERT (Reimers & Gurevych, 2019), and one using $\text{LM}_{\text{SR}}$, a pretrained LM for question-relation matching over the KG (Zhang et al., 2022a). Our experimental results suggest that, although these GNNs retrieve different KG information, they both improve RAG-based KGQA.

## 4.2 GNN Implementation

**Classification layer**: After $L$ GNN layers, we obtain node representation matrix $\boldsymbol{H}^{(L)} \in \mathbb{R}^{|\mathcal{V}| \times d}$. To perform classification, we obtain the node probability matrix $P = \text{softmax}(\boldsymbol{H}^{(L)}\boldsymbol{W})$, where $\boldsymbol{W} \in R^{d \times 1}$ is a learnable projection layer followed by $\text{softmax}$ normalization. Answer nodes should have larger probability $p_v \in [0, 1]$ than non-answer nodes.

**Node and relation embeddings**: We use pretrained models, such as SBERT or other LMs, to encode relation embeddings. We obtain node embeddings by aggregating the adjacent relation embeddings of nodes, which has been shown to generalize better to new entities (He et al., 2021; Choi et al., 2024). The formula is $\boldsymbol{h}_v^{(0)} = ReLU(\sum_{\boldsymbol{r} \in N_r(v)} \boldsymbol{W}_r \boldsymbol{r})$, where $\boldsymbol{r}$ is the relation embedding and $\boldsymbol{W}$ is learnable. During training, we optimize the GNN parameters, but not the relation embeddings obtained via the pretrained models.

**Question Representations**: As complex questions might consist of multiple subquestions, we obtain $K$ question representations to better capture different question parts (Qiu et al., 2020), as shown in Equation 3. To capture multiple question's contexts, each question representation $\boldsymbol{q}_k \in \mathbb{R}^d, k \in K$, is initialized by dynamically attending to different question's tokens. ). First, we derive a representation $\boldsymbol{q}_j \in \mathbb{R}^d$ for each token $j$ of the question and a question representation, e.g., via CLS pooling, $\boldsymbol{q}_c \in \mathbb{R}^d$ with pre-trained language models, such as SBERT. Equation 3 becomes

$$\boldsymbol{q}_k = \gamma_k(\text{LM}(q)) = \sum_j a_{k,j} \boldsymbol{q}_j, \tag{4}$$

where $j$ denotes is the $j$-th token position and $a_{k,j} \in [0, 1]$ is an attention weight. At each iteration $k$, weight $a_{k,j}$ is dynamically adjusted by encouraging attention to new question parts via:

$$a_{k,j} = \text{softmax}_j(\boldsymbol{W}_a(\tilde{\boldsymbol{q}}_k \odot \boldsymbol{q}_j) \tag{5}$$

$$\tilde{\boldsymbol{q}}_k = \boldsymbol{W}_k(\boldsymbol{q}_{k-1} || \boldsymbol{q}_c || \boldsymbol{q}_{k-1} \odot \boldsymbol{q}_c || \boldsymbol{q}_c - \boldsymbol{q}_{k-1}), \tag{6}$$

where $\boldsymbol{W}_a \in \mathbb{R}^{d \times d}$ and $\boldsymbol{W}_k \in \mathbb{R}^{d \times 4d}$ are learnable parameters.

## 4.3 RAG with LLM

In text RAG, retrieval is performed on a document corpus $\mathcal{D}$ (Section 3-LLM RAG). In KGQA, the corpus is the node set $\mathcal{V}$. GNN-RAG uses the GNN model as the scoring model to obtain the top relevant nodes for answering the query, $[v_1, \ldots, v_k] = \text{top-}k_{v_i \in \mathcal{V}}\text{GNN}(v_i, q)$. In order to provide more context to the LLM, we extract the shortest paths between question entities and the GNN top scored nodes. After obtaining the reasoning paths by GNN-RAG, we verbalize them and give them as input to a downstream LLM, such as ChatGPT or LLaMA. However, LLMs are sensitive to the input prompt template and the way that the graph information is verbalized.

To alleviate this issue, we opt to follow RAG prompt tuning (Lin et al., 2023; Zhang et al., 2024b) for LLMs that have open weights and are feasible to train. A LLaMA2-Chat-7B model is fine-tuned based on the training question-answer pairs to generate a list of correct answers, given the prompt: "`Based on the reasoning paths, please answer the given question.\n Reasoning Paths: {Reasoning Paths} \n Question: {Question}`".

The reasoning paths are verbalized as "{question entity} → {relation} → {entity} → ⋯ → {relation} → {answer entity} \n" (see Figure 4). During training, the reasoning paths are the shortest paths from question entities to answer entities. During inference, the reasoning paths are obtained by GNN-RAG.

### 4.4 RETRIEVAL STUDY: WHY GNNs & THEIR LIMITATIONS

GNNs leverage the graph structure to retrieve relevant parts of the KG that contain multi-hop information. We provide experimental evidence on why GNNs are good retrievers for multi-hop KGQA. We train two different GNNs, a deep one ($L = 3$) and a shallow one ($L = 1$), and measure their retrieval capabilities. We report

Table 1: Retrieval results for WebQSP.

| Retriever | 1-hop questions | | 2-hop questions | |
|---|---|---|---|---|
| | #Input Tok. | %Ans. Cov. | #Input Tok. | %Ans. Cov. |
| RoG (Luo et al., 2024) | 150 | **87.1** | 435 | 82.1 |
| GNN ($L = 1$) | 112 | 83.6 | 2,582 | 79.8 |
| GNN ($L = 3$) | 105 | 82.4 | 357 | **88.5** |

the 'Answer Coverage' metric, which evaluates whether the retriever is able to fetch at least one correct answer for RAG. Note that 'Answer Coverage' does not measure downstream KGQA performance but whether the retriever fetches relevant KG information. '#Input Tokens' denotes the median number of the input tokens of the retrieved KG paths. Table 1 shows GNN retrieval results for single-hop and multi-hop questions of the WebQSP dataset compared to an LLM-based retriever (RoG; Equation 2). The results indicate that deep GNNs ($L = 3$) can handle the complex graph structure and retrieve useful *multi-hop* information more effectively (%Ans. Cov.) and efficiently (#Input Tok.) than the LLM and the shallow GNN.

On the other hand, the limitation of GNNs is for simple (1-hop) questions, where accurate question-relation matching is more important than deep graph search (see our Theorem in Appendix B that states this GNN limitation). In such cases, the LLM retriever is better at selecting the right KG information due to its natural language understanding abilities (we provide an example later in Figure 6).

### 4.5 RETRIEVAL AUGMENTATION (RA)

Retrieval augmentation (RA) combines the retrieved KG information from different approaches to increase diversity and answer recall. Motivated by the results in Section 4.4, we present a RA technique (**GNN-RAG+RA**), which complements the GNN retriever with an LLM-based retriever to combine their strengths on multi-hop and single-hop questions, respectively. Specifically, we experiment with the RoG retrieval, which is described in Equation 2. During inference, we take the union of the reasoning paths retrieved by the two retrievers.

A downside of LLM-based retrieval is that it requires multiple generations (beam-search decoding) to retrieve diverse paths, which trades efficiency for effectiveness (we provide a performance analysis in Appendix B). A cheaper alternative is to perform RA by combining the outputs of different GNNs, which are equipped with different LMs in Equation 3. Our **GNN-RAG+Ensemble** combines two different GNNs (GNN+SBERT & GNN+LM$_{SR}$) as input for RA.

## 5 EXPERIMENTAL SETUP

**KGQA Datasets**. We experiment with widely used KGQA benchmarks: WebQuestionsSP (We-bQSP) (Yih et al., 2015), Complex WebQuestions 1.1 (CWQ) (Talmor & Berant, 2018), and MetaQA-3 Zhang et al. (2018). **WebQSP** contains 4,737 natural language questions that are answerable using a subset Freebase KG (Bollacker et al., 2008). The questions require up to 2-hop reasoning within this KG. **CWQ** contains 34,699 total complex questions that require up to 4-hops of reasoning over the KG. **MetaQA-3** consists of 3-hop questions in the domain of WikiMovies Miller et al. (2016). We provide the detailed dataset statistics in Appendix D.

**Implementation & Evaluation**. For subgraph retrieval, we use the linked entities and the pagerank algorithm to extract dense graph information (He et al., 2021). We employ ReaRev (Mavromatis & Karypis, 2022), which is a GNN targeting at *deep* KG reasoning (Section 4.4), for GNN-RAG. The default implementation is to combine ReaRev with SBERT as the LM in Equation 3. In addition, we combine ReaRev with LM$_{SR}$, which is obtained by following the implementation of SR (Zhang et al.,

Table 2: Performance comparison of different methods on the two KGQA benchmarks. We denote the **best** and second-best method. Hit is used for LLM evaluation due to their free-form generation and H@1/F1 metrics are used for methods that return a list of scored answers.

| Type | Method | WebQSP | | | CWQ | | |
|---|---|---|---|---|---|---|---|
| | | Hit | H@1 | F1 | Hit | H@1 | F1 |
| Embedding | KV-Mem Miller et al. (2016) | – | 46.7 | 38.6 | – | 21.1 | – |
| | EmbedKGQA Saxena et al. (2020) | – | 66.6 | – | – | – | – |
| | TransferNet Shi et al. (2021) | – | 71.4 | – | – | 48.6 | – |
| | Rigel Sen et al. (2021) | – | 73.3 | – | – | 48.7 | – |
| GNN | GraftNet Sun et al. (2018) | – | 66.7 | 62.4 | – | 36.8 | 32.7 |
| | PullNet Sun et al. (2019) | – | 68.1 | – | – | 45.9 | – |
| | NSM He et al. (2021) | – | 68.7 | 62.8 | – | 47.6 | 42.4 |
| | SR+NSM(+E2E) (Zhang et al., 2022a) | – | 69.5 | 64.1 | – | 50.2 | 47.1 |
| | NSM+h He et al. (2021) | – | 74.3 | 67.4 | – | 48.8 | 44.0 |
| | SQALER Atzeni et al. (2021) | – | 76.1 | – | – | – | – |
| | UniKGQA (Jiang et al., 2023b) | – | 77.2 | 72.2 | – | 51.2 | 49.1 |
| | ReaRev (Mavromatis & Karypis, 2022) | – | 76.4 | 70.9 | – | 52.9 | 47.8 |
| | ReaRev + LM$_{SR}$ | – | 77.5 | 72.8 | – | 53.3 | 49.7 |
| LLM | Flan-T5-xl (Chung et al., 2024) | 31.0 | – | – | 14.7 | – | – |
| | Alpaca-7B (Taori et al., 2023) | 51.8 | – | – | 27.4 | – | – |
| | LLaMA2-Chat-7B (Touvron et al., 2023) | 64.4 | – | – | 34.6 | – | – |
| | ChatGPT | 66.8 | – | – | 39.9 | – | – |
| | ChatGPT+CoT | 75.6 | – | – | 48.9 | – | – |
| KG+LLM | KAPING (Baek et al., 2023) | 73.9 | – | – | – | – | – |
| | KD-CoT (Wang et al., 2023) | 68.6 | – | 52.5 | 55.7 | – | – |
| | StructGPT (Jiang et al., 2023a) | 72.6 | – | – | – | – | – |
| | KB-BINDER (Li et al., 2023) | 74.4 | – | – | – | – | – |
| | ToG+LLaMA2-70B (Sun et al., 2024) | 68.9 | – | – | 57.6 | – | – |
| | ToG+ChatGPT (Sun et al., 2024) | 76.2 | – | – | 58.9 | – | – |
| | ToG+GPT-4 (Sun et al., 2024) | 82.6 | – | – | **69.5** | – | – |
| | RoG (Luo et al., 2024) | 85.7 | 80.0 | 70.8 | 62.6 | 57.8 | 56.2 |
| GNN+LLM | G-Retriever (He et al., 2024) | – | 70.1 | – | – | – | – |
| | **GNN-RAG** | 85.7 | 80.6 | 71.3 | 66.8 | 61.7 | 59.4 |
| | **GNN-RAG+RA** | **90.7** | **82.8** | **73.5** | 68.7 | 62.8 | 60.4 |

We use the default GNN-RAG (+RA) implementation. GNN-RAG, RoG, KD-CoT, and G-Retriever use 7B fine-tuned LLaMA2 models. KD-CoT employs ChatGPT as well.

2022a). We employ RoG (Luo et al., 2024) for RAG-based prompt tuning (Section 4.3). For KGQA evaluation, we adopt Hit, Hits@1 (H@1), and F1 metrics. Hit measures if any of the true answers is found in the generated response, which is typically employed when evaluating LLMs. H@1 is the accuracy of the top/first predicted answer. F1 takes into account the recall (number of true answers found) and the precision (number of false answers found) of the generated answers, making it a more faithful metric. For retrieval evaluation, we use Hit@$k$, which evaluates whether a correct answer is retrieved in the top-$k$ retrieved nodes. Further experimental setup details are provided in Appendix D.

**Competing Methods**. We compare with SOTA GNN and LLM methods for KGQA (Mavromatis & Karypis, 2022; Li et al., 2023). We also include earlier embedding-based methods (Saxena et al., 2020) as well as zero-shot/few-shot LLMs (Taori et al., 2023). We do not compare with semantic parsing methods (Yu et al., 2022; Gu et al., 2023) as they use additional training data (SPARQL annotations), which are difficult to obtain in practice. Furthermore, we compare GNN-RAG with LLM-based retrieval approaches (Luo et al., 2024; Sun et al., 2024) in terms of efficiency and effectiveness.

# 6 RESULTS

**Main Results**. Table 2 presents performance results of different KGQA methods. GNN-RAG is the method that performs overall the best, achieving state-of-the-art results on the two KGQA benchmarks in almost all metrics. The results show that equipping LLMs with GNN-based retrieval boosts their reasoning ability significantly (GNN+LLM vs. KG+LLM). Specifically, GNN-RAG+RA outperforms RoG by 5.0–6.1% points at Hit, while it outperforms or matches ToG+GPT-4 performance, using an LLM with only 7B parameters and much fewer LLM calls – we estimate ToG+GPT-4 has an overall cost above $800, while GNN-RAG can be deployed on a single 24GB GPU. GNN-RAG+RA

Table 3: Performance analysis on multi-hop (hops$\geq 2$) and multi-entity (entities$\geq 2$) questions.

| Method | WebQSP (F1) | | CWQ (F1) | | MetaQA-3 (H@1) |
|---|---|---|---|---|---|
| | multi-hop | multi-entity | multi-hop | multi-entity | multi-hop |
| LLM (No RAG) | 48.4 | 61.5 | 33.7 | 32.3 | 29.7 |
| GNN | 58.8 | 70.4 | 57.7 | 54.2 | 98.6 |
| RoG | 63.3 | 65.1 | 59.3 | 58.3 | 84.8 |
| GNN-RAG | 69.8 | 82.3 | 68.2 | 64.8 | 98.6 |
| GNN-RAG+RA | 71.1 | 88.8 | 69.3 | 65.6 | 98.6 |
| Absolute Improv. | **+7.8** | **+23.7** | **+10.0** | **+7.3** | **+13.8** |

Table 4: Performance comparison (F1 at KGQA) of different retrieval augmentations (Section 4.5). '#LLM Calls' are controlled by the hyperparameter $k$ (number of beams) during beam-search decoding for LLM-based retrievers, '#Input Tokens' denotes the median number of tokens.

| Retriever | #LLM Calls | Retrieval Metrics | | | KGQA Performance |
|---|---|---|---|---|---|
| | | #Input Tokens | Hit@1 (%) | Hit@10 (%) | F1 (%) |
| | | WebQSP / CWQ | WebQSP / CWQ | WebQSP / CWQ | WebQSP / CWQ |
| (a) RoG | 3 | 202 / 325 | 59.9 / 25.9 | 78.1 / 54.5 | 70.8 / 56.2 |
| (b) GNN-RAG | 0 | 144 / 207 | 76.4 / 52.9 | 82.6 / 64.1 | 71.3 / 59.4 |
| (c) GNN-RAG+RA | 3 | 299 / 540 | 76.4 / 52.9 | 89.9 / 71.1 | 73.5 / 60.4 |
| (d) GNN-RAG+Ensemble | 0 | 156 / 281 | 77.5 / 53.3 | 84.7 / 66.7 | 71.7 / 57.5 |
| (e) GNN | 0 | – | 76.4 / 52.9 | 82.6 / 64.1 | 70.9 / 47.8 |

outperforms ToG+ChatGPT by up to 14.5% points at Hit and the best performing GNN by 5.3–9.5% points at Hits@1 and by 0.7–10.7% points at F1.

**Complex KGQA**. Table 3 compares complex KGQA performance results on multi-hop questions, where answers are more than one hop away from the question entities, and multi-entity questions, which have more than one question entities. GNN-RAG leverages GNNs to handle complex graph information and outperforms RoG (LLM-based retrieval) by 6.5–17.2% points at F1 on WebQSP, by 8.5–8.9% points at F1 on CWQ, and by 13.8% points at H@1 on MetaQA-3. In addition, GNN-RAG+RA offers an additional improvement by up to 6.5% points at F1. The results show that GNN-RAG is an effective retrieval method when the questions involve complex graph information.

Table 5: Performance comparison of different graph retrievers in RAG for KGQA.

| Retriever | WebQSP | | | CWQ | | |
|---|---|---|---|---|---|---|
| | Hit | H@1 | F1 | Hit | H@1 | F1 |
| Dense Subgraph | 70.2 | 68.7 | 54.3 | 47.1 | 45.5 | 41.9 |
| GNN-RAG: GraftNet | 82.8 | 78.6 | 69.8 | 58.2 | 51.9 | 49.4 |
| GNN-RAG: NSM | 85.0 | 79.6 | 70.4 | 58.5 | 52.5 | 50.1 |
| GNN-RAG: ReaRev | **85.7** | **80.6** | **71.3** | **66.8** | **61.7** | **59.4** |

**Retrieval Assessment**. Table 4 assesses retrieval perfomance of different graph retrieval approaches, along with donwstream KGQA perfomance. Based on the results, we make the following conclusions:

1. GNN-based retrieval is more efficient (#LLM Calls, #Input Tokens) and effective (F1) than LLM-based retrieval (RoG), especially for complex questions (CWQ); see rows (a) vs. (b).

2. GNN-based retrieval achieves remarkable performance, outperforming LLM-based retrieval by 17.6–27.4% points at H@1; e.g., see rows (a) vs. (d)/(e).

3. Retrieval augmentation works the best (Hit@$k$ and KGQA F1) when combining GNN-induced reasoning paths with LLM-induced reasoning paths as they fetch non-overlapping KG information (increased #Input Tokens) that improves retrieval for KGQA; see row (c).

4. Augmenting all retrieval approaches does not necessarily cause improved performance (F1) as the long input (#Input Tokens) may confuse the LLM; see row (d) at CWQ.

Although GNN-RAG outperforms LLM-based retrieval, *we note that weak GNNs are not effective retrievers*. GNN-RAG employs ReaRev as its GNN retriever, which is a powerful GNN for deep KG reasoning. In Table 5, we ablate on the impact of the GNN used for retrieval, i.e., how strong and weak GNNs affect KGQA performance. We experiment with GraftNet and NSM GNNs, which are less powerful than ReaRev at KGQA. The results are presented in Table5 and show that strong GNNs

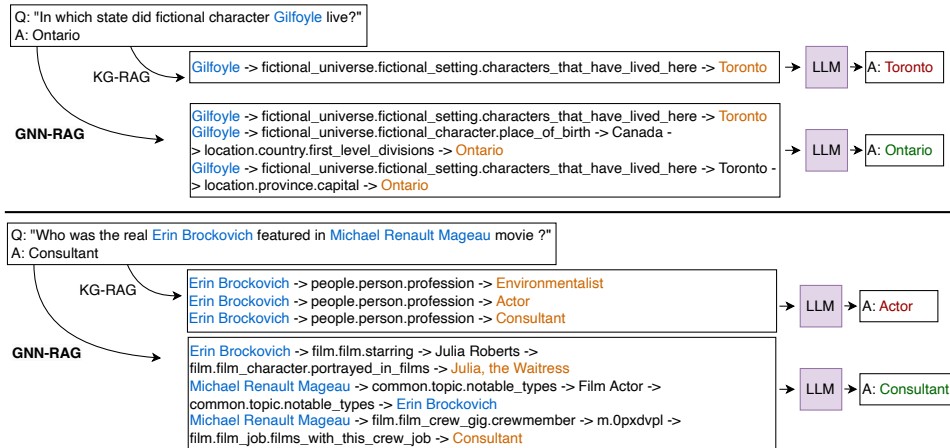

Figure 5: Two case studies that illustrate how GNN-RAG improves the LLM's faithfulness. In both cases, GNN-RAG retrieves *multi-hop* information that is necessary for answering the complex questions.

(ReaRev) are essential for state-of-the-art KGQA performance. Although retrieval with weak GNNs (NSM and GraftNet) still outperforms dense subgraph retrieval, it performs worse than strong GNNs by up to 9.8% at H@1.

**Retrieval Effect on LLMs**. Table 6 presents performance results of various LLMs using GNN-RAG or LLM-based retrievers (RoG and ToG). We report the Hit metric as it is difficult to extract the number of answers from LLM's output. GNN-RAG (+RA) is the retrieval approach that achieves the largest improvements for RAG. For instance, GNN-RAG+RA improves ChatGPT by up to 6.5% points at Hit over RoG and ToG. Moreover, GNN-RAG substantially improves the KGQA performance of weaker LLMs, such as Alpaca-7B and Flan-T5-xl. The improvement over RoG is up to 13.2% points at Hit, while GNN-RAG outperforms LLaMA2-Chat-70B+ToG using a lightweight 7B LLaMA2 model. The results demonstrate that GNN-RAG can be integrated with other LLMs to improve their KGQA reasoning without retraining.

**Case Studies on Faithfulness**. Figure 5 illustrates two case studies from the CWQ dataset, showing how GNN-RAG improves LLM's faithfulness, i.e., how well the LLM follows the question's instructions and uses the right information from the KG. We provide additional discussions on Appendix A.

Further ablation studies are provided in Appendix E. Limitations are discussed in Appendix F.

Table 6: Retrieval effect on performance (% Hit) using various LLMs.

| Method | WebQSP | CWQ |
|---|---|---|
| ChatGPT | 51.8 | 39.9 |
| + ToG | 76.2 | 58.9 |
| + RoG | 81.5 | 52.7 |
| + GNN-RAG (+RA) | 85.3 (**87.9**) | 64.1 (**65.4**) |
| Alpaca-7B | 51.8 | 27.4 |
| + RoG | 73.6 | 44.0 |
| + GNN-RAG (+RA) | 76.2 (**76.5**) | **54.5** (50.8) |
| LLaMA2-Chat-7B | 64.4 | 34.6 |
| + RoG | 84.8 | 56.4 |
| + GNN-RAG (+RA) | 85.2 (**88.5**) | 62.7 (**62.9**) |
| LLaMA2-Chat-70B | 57.4 | 39.1 |
| + ToG | 68.9 | 57.6 |
| Flan-T5-xl | 31.0 | 14.7 |
| + RoG | 67.9 | 37.8 |
| + GNN-RAG (+RA) | **74.5** (72.3) | **51.0** (41.5) |

## 7 CONCLUSION

We introduce GNN-RAG, a novel graph neural method for enhancing RAG in KGQA with GNNs. Our **contributions** are the following. (1) **Framework**: GNN-RAG tailors GNNs for KG retrieval due to their ability to handle complex graph information. Similar to retrieval in text-based RAG, GNN-RAG can be seamlessly integrated with different downstream LLMs. (2) **Effectiveness & Faithfulness**: GNN-RAG achieves state-of-the-art performance in two widely used KGQA benchmarks (WebQSP and CWQ). Furthermore, GNN-RAG is shown to retrieve multi-hop information that is necessary for faithful LLM reasoning on complex questions. (3) **Efficiency**: GNN-RAG improves vanilla LLMs on KGQA performance without incurring additional LLM calls as existing RAG systems for KGQA require. In addition, GNN-RAG outperforms or matches GPT-4 performance with a 7B tuned LLM.

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

# Appendix / supplemental material

## A CASE STUDIES ON FAITHFULNESS

Figure 5 illustrates two case studies from the CWQ dataset, showing how GNN-RAG improves LLM's faithfulness, i.e., how well the LLM follows the question's instructions and uses the right information from the KG.

In both cases, GNN-RAG retrieves multi-hop information, which is necessary for answering the questions correctly. In the first case, GNN-RAG retrieves both crucial facts `<Gilfoyle → characters_that_have_lived_here → Toronto>` and `<Toronto → province.capital → Ontario>` that are required to answer the question, unlike the KG-RAG baseline (RoG) that fetches only the first fact. In the second case, the KG-RAG baseline incorrectly retrieves information about `<Erin Brockovich → person>` and not `<Erin Brockovich → film_character>` that the question refers to. GNN-RAG uses GNNs to explore how `<Erin Brockovich>` and `<Michael Renault Mageau>` entities are related in the KG, resulting into retrieving facts about `<Erin Brockovich → film_character>`. The retrieved facts include important information `<films_with_this_crew_job → Consultant>`.

Figure 6 illustrates one case study from the WebQSP dataset, showing how RA (Section 4.5) improves GNN-RAG. Initially, the GNN does not retrieve helpful information due to its limitation to understand natural language, i.e., that `<jurisdiction.bodies>` usually "`make the laws`". GNN-RAG+RA retrieves the right information, helping the LLM answer the question correctly.

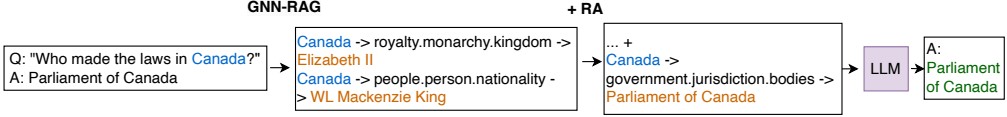

Figure 6: One case study that illustrates the benefit of retrieval augmentation (RA). RA uses LLMs to fetch semantically relevant KG information, which may have been missed by the GNN.

## B ANALYSIS

In this section, we analyze the reasoning and retrieval abilities of GNN and LLMs, respectively.

**Definition B.1** (Ground-truth Subgraph). Given a question $q$, we define its ground-truth reasoning subgraph $\mathcal{G}_q^*$ as the union of the ground-truth reasoning paths that lead to the correct answers $\{a\}$. Reasoning paths are defined as the KG paths that reach the answer nodes, starting from the question entities $\{e\}$, e.g., `<Jamaica → language_spoken → English>` for question "`Which language do Jamaican people speak?`". In essence, $\mathcal{G}_q^*$ contains only the necessary entities and relations that are needed to answer $q$.

Table 7: Efficiency vs. effectiveness trade-off of LLM-based retrieval.

| Retrieval | #LLM Calls (efficiency) | Answer Hit (%) (effectiveness) |
|---|---|---|
| RoG (Luo et al., 2024) | 3 | 85.7 |
| | 1 | 77.2 |
| ToG (Sun et al., 2024) | up to 21 | 76.2 |
| | 3 | 66.3 |
| GNN-RAG | 0 | 87.2 |

#LLM Calls are controlled by the hyperparameter $k$ (number of beams) during beam-search decoding.

**Definition B.2** (Effective Reasoning). We define that a model $M$ *reasons effectively* if its output is $\{a\} = M(\mathcal{G}_q^*, q)$, i.e., the model returns the correct answers given the ground-truth subgraph $\mathcal{G}_q^*$.

As KGQA methods do not use the ground-truth subgraph $\mathcal{G}_q^*$ for reasoning, but the retrieved subgraph $\mathcal{G}_q$, we identify two cases in which the reasoning model *cannot* reason effectively, i.e., $\{a\} \neq M(\mathcal{G}_q, q)$.

Case 1: $\mathcal{G}_q \subset \mathcal{G}_q^*$, i.e., the retrieved subgraph $\mathcal{G}_q$ does not contain all the necessary information for answering $q$. An application of this case is when we use LLMs for retrieval. As LLMs are not designed to handle complex graph information, the retrieved subgraph $\mathcal{G}_q$ may contain incomplete

KG information. Existing LLM-based methods rely on employing an increased number of LLM calls (beam search decoding) to fetch diverse reasoning paths that approximate $\mathcal{G}_q^*$. Table 7 provides experimental evidence that shows how LLM-based retrieval trades computational efficiency for effectiveness. *In particular, when we switch from beam-search decoding to greedy decoding for faster LLM retrieval, the KGQA performance drops by 8.3–9.9% points at answer hit.*

Case 2: $\mathcal{G}_q^* \subset \mathcal{G}_q$ and model $M$ cannot "filter-out" irrelevant facts during reasoning. An application of this case is when we use GNNs for reasoning. GNNs cannot understand the textual semantics of KGs and natural questions the same way as LLMs do, and they reason ineffectively if they cannot tell the irrelevant KG information. We develop the following Theorem that supports this case for GNNs.

**Theorem B.3** (Simplified). *Under mild assumptions and due to the sum operator of GNNs in Equation 1, a GNN can reason effectively by selecting question-relevant facts and filtering-out question-irrelevant facts through $\omega(q, r)$.*

We provide the full theorem and its proof in Appendix C. Theorem B.3 suggests that GNNs need to perform semantic matching via function $\omega(q, r)$ apart from leveraging the graph information encoded in the KG. Our analysis suggests that GNNs lack reasoning abilities for KGQA if they cannot perform effective semantic matching between the KG and the question.

## C    FULL THEOREM & PROOF

To analyze under which conditions GNN perform well for KGQA, we use the ground-truth subgraph $\mathcal{G}_q^*$ for a question $q$, as defined in Definition B.1. We compare the output representations of a GNN over the ground-truth $\mathcal{G}_q^*$ and another $\mathcal{G}_q$ to measure how close the two outputs are.

We always assume $\mathcal{G}_q^* \subseteq \mathcal{G}_q$ for a question $q$. 1-hop facts that contain $v$ are denoted as $\mathcal{N}_v^*$.

**Definition C.1.** Let $M$ be a GNN model for answering question $q$ over a KG $\mathcal{G}_q$, where the output is computed by $M(q, \mathcal{G}_q)$. $M$ consists of $L$ reasoning steps (GNN layers). We assume $M$ is an effective reasoner, according to Definition B.2. Furthermore, we define the reasoning process $\mathcal{R}_{M,q,\mathcal{G}_q}$ as the sequence of the derived node representations at each step $l$, i.e.,

$$\mathcal{R}_{M,q,\mathcal{G}_q} = \left\{ \{\boldsymbol{h}_v^{(1)} : v \in \mathcal{G}_q\}, \ldots, \{\boldsymbol{h}_v^{(L)} : v \in \mathcal{G}_q\} \right\}. \tag{7}$$

We also define the optimal reasoning process for answering question $q$ with GNN $M$ as $\mathcal{R}_{M,q,\mathcal{G}_q^*}$. We assume that zero node representations do not contribute in Equation 7.

**Lemma C.2.** *If two subgraphs $\mathcal{G}_1$ and $\mathcal{G}_2$ have the same nodes, and a GNN outputs the same node representations for all nodes $v \in \mathcal{G}_1$ and $v \in \mathcal{G}_2$ at each step $l$, then the reasoning processes $\mathcal{R}_{M,q,\mathcal{G}_1}$ and $\mathcal{R}_{M,q,\mathcal{G}_2}$ are identical.*

This is true as $\boldsymbol{h}_v^{(l)}$ with $l = 1, \ldots, L$ for both $\mathcal{G}_1$ and $\mathcal{G}_2$ and by using Definition C.1 to show $\mathcal{R}_{M,q,\mathcal{G}_1} = \mathcal{R}_{M,q,\mathcal{G}_2}$. Note that Lemma C.2 does not make any assumptions about the actual edges of $\mathcal{G}_1$ and $\mathcal{G}_2$.

To analyze the importance of semantic matching for GNNs, we consider the following GNN update

$$\boldsymbol{h}_v^{(l)} = \psi\Big(\boldsymbol{h}_v^{(l-1)}, \sum_{v' \in \mathcal{N}_v} \omega(q, r) \cdot \boldsymbol{m}_{vv'}^{(l)}\Big). \tag{8}$$

where $\omega(\cdot, \cdot) : \mathbb{R}^d \times \mathbb{R}^d \to \{0, 1\}$ is a binary function that decides if fact $(v, r, v')$ is relevant to question $q$ or not. In practice, $\omega$ is implemented by LMs (Reimers & Gurevych, 2019). Neighbor messages $\boldsymbol{m}_{vv'}^{(l)}$ are aggregated by a sum-operator, which is typically employed in GNNs. Function $\psi(\cdot)$ combines representations among consecutive GNN layers. We assume $\boldsymbol{h}_v^{(0)} \in \mathbb{R}^d$ and that $\psi\Big(h_v^{(0)}, 0^d\Big) = 0^d$

**Theorem C.3.** *If $\omega(q, r) = 0, \forall(v, r, v') \notin \mathcal{G}_q^*$ and $\omega(q, r) = 1, \forall(v, r, v') \in \mathcal{G}_q^*$, then $\mathcal{R}_{M,q,\mathcal{G}_q}$ is an optimal reasoning process of GNN $M$ for answering $q$.*

*Proof.* We show that

$$\sum_{v' \in \mathcal{N}_v} \omega(q, r) \cdot \boldsymbol{m}_{vv'}^{(l)} = \sum_{v' \in \mathcal{N}_v^*} \boldsymbol{m}_{vv'}^{(l)}, \tag{9}$$

which gives that $\mathcal{R}_{M,q,\mathcal{G}_q} = \mathcal{R}_{M,q,\mathcal{G}_q^*}$ via Lemma C.2. This is true if

$$\omega(q,r) = \begin{cases} 1 & \text{if } (v,r,v') \in \mathcal{N}_v^*, \\ 0 & \text{if } (v,r,v') \notin \mathcal{N}_v^*, \end{cases} \tag{10}$$

which means that GNNs need to filter-out question irrelevant facts. We consider two cases.

**Case 1**. Let $u$ denote a node that is present in $\mathcal{G}_q$, but not in $\mathcal{G}_q^*$. Then, all facts that contain $u$ are not present in $\mathcal{G}_q^*$. Condition $\omega(q,r) = 0, \forall (v,r,v') \notin \mathcal{G}_q^*$ of Theorem C.3 gives that

$$\omega(q,r) = 0, \forall (u,r,v'), \text{ and}$$
$$\omega(q,r) = 0, \forall (v,r,u). \tag{11}$$

as node $u \notin \mathcal{G}_q^*$. This gives

$$\sum_{v' \in \mathcal{N}(u)} \omega(q,r) \cdot \boldsymbol{m}_{uv'}^{(l)} = 0, \tag{12}$$

as no edges will contribute to the GNN update. With $\psi\left(h_v^{(0)}, 0^d\right) = 0^d$, we have

$$h_u^{(l)} = 0^d, \forall u \notin \mathcal{G}_q^* \text{ with } l = \{1, \ldots, L\}, \tag{13}$$

which means that nodes $u \notin \mathcal{G}_q^*$ do not contribute to the reasoning process $\mathcal{R}_{M,q,\mathcal{G}_q}$; see Definition C.1.

**Case 2**. Let $p$ denote a relation between two nodes $v$ and $v'$ that is present in $\mathcal{G}_q$, but not in $\mathcal{G}_q^*$. We decompose the GNN update to

$$\sum_{v' \in \mathcal{N}_r(v)} \omega(q,r) \cdot \boldsymbol{m}_{vv'}^{(l)} + \sum_{v' \in \mathcal{N}_p(v)} \omega(q,p) \cdot \boldsymbol{m}_{vv'}^{(l)}, \tag{14}$$

where the first term includes facts $\mathcal{N}_r$ that are present in $\mathcal{G}_q^*$ and the second term includes facts $\mathcal{N}_p$ that are present in $\mathcal{G}_q$ only. Using the condition $\omega(q,r) = 0, \forall (v,r,v') \notin \mathcal{G}_q^*$ of Theorem C.3, we have

$$\sum_{v' \in \mathcal{N}_p(v)} \omega(q,p) \cdot \boldsymbol{m}_{vv'}^{(l)} = 0. \tag{15}$$

Using condition $\omega(q,r) = 1, \forall (v,r,v') \in \mathcal{G}_q^*$, we have

$$\sum_{v' \in \mathcal{N}_r(v)} \omega(q,r) \cdot \boldsymbol{m}_{vv'}^{(l)} = \sum_{v' \in \mathcal{N}_r(v)} \boldsymbol{m}_{vv'}^{(l)}. \tag{16}$$

Combining the two above expression gives

$$\sum_{v' \in \mathcal{N}_v} \omega(q,r) \cdot \boldsymbol{m}_{vv'}^{(l)} = \sum_{v' \in \mathcal{N}_r(v)} \boldsymbol{m}_{vv'}^{(l)} = \sum_{v' \in \mathcal{N}_v^*} \boldsymbol{m}_{vv'}^{(l)}. \tag{17}$$

It is straightforward to obtain $\mathcal{R}_{M,q,\mathcal{G}_q} = \mathcal{R}_{M,q,\mathcal{G}_q^*}$ via Lemma C.2 in this case.

**Putting it altogether**. Combining Case 1 and Case 2, nodes $u \notin \mathcal{G}_q^*$ do not contribute to $\mathcal{R}_{M,q,\mathcal{G}_q}$, while for other nodes we have $\mathcal{R}_{M,q,\mathcal{G}_q} = \mathcal{R}_{M,q,\mathcal{G}_q^*}$. Thus, overall we have $\mathcal{R}_{M,q,\mathcal{G}_q} = \mathcal{R}_{M,q,\mathcal{G}_q^*}$. □

## D EXPERIMENTAL SETUP

**KGQA Datasets**. We experiment with two widely used KGQA benchmarks: WebQuestionsSP (WebQSP) Yih et al. (2015), Complex WebQuestions 1.1 (CWQ) Talmor & Berant (2018). We also experiment with MetaQA-3 Zhang et al. (2018) dataset. We provide the dataset statistics Table 8. **WebQSP** contains 4,737 natural language questions that are answerable using a subset Freebase KG (Bollacker et al., 2008). This KG contains 164.6 million facts and 24.9 million entities. The questions require up to 2-hop reasoning within this KG. Specifically, the model needs to aggregate over two KG facts for 30% of the questions, to reason over constraints for 7% of the questions, and to use a single KG fact for the rest of the questions. **CWQ** is generated from WebQSP by extending the question entities or adding constraints to answers, in order to construct more complex multi-hop

Table 8: Datasets statistics. "avg.$|\mathcal{V}_q|$" denotes average number of entities in subgraph, and "coverage" denotes the ratio of at least one answer in subgraph.

| Datasets | Train | Dev | Test | avg. $|\mathcal{V}_q|$ | coverage (%) |
|---|---|---|---|---|---|
| WebQSP | 2,848 | 250 | 1,639 | 1,429.8 | 94.9 |
| CWQ | 27,639 | 3,519 | 3,531 | 1,305.8 | 79.3 |
| MetaQA-3 | 114,196 | 14,274 | 14,274 | 497.9 | 99.0 |

questions (34,689 in total). There are four types of questions: composition (45%), conjunction (45%), comparative (5%), and superlative (5%). The questions require up to 4-hops of reasoning over the KG, which is the same KG as in WebQSP. **MetaQA-3** consists of more than 100k 3-hop questions in the domain of movies. The questions were constructed using the KG provided by the WikiMovies Miller et al. (2016) dataset, with about 43k entities and 135k triples. For MetaQA-3, we use 1,000 (1%) of the training questions.

**Implementation**. For subgraph retrieval, we use the linked entities to the KG provided by Yih et al. (2015) for WebQSP, by Talmor & Berant (2018) for CWQ. We obtain dense subgraphs by He et al. (2021). It runs the PageRank Nibble Andersen et al. (2006) (PRN) method starting from the linked entities to select the top-$m$ ($m = 2,000$) entities to be included in the subgraph.

We employ ReaRev[1] (Mavromatis & Karypis, 2022) for GNN reasoning (Section 4.1) and RoG[2] (Luo et al., 2024) for RAG-based prompt tuning (Section 4.3), following their official implementation codes. In addition, we empower ReaRev with $LM_{SR}$ (Section 4.1), which is obtained by following the implementation of SR[3] (Zhang et al., 2022a). For both training and inference of these methods, we use their suggested hyperparameters, without performing further hyperparameter search. Model selection is performed based on the validation data. Experiments with GNNs were performed on a Nvidia Geforce RTX-3090 GPU over 128GB RAM machine. Experiments with LLMs were performed on 4 A100 GPUs connected via NVLink and 512 GB of memory. The experiments are implemented with PyTorch.

For LLM prompting during retrieval (Section 4.5), we use the following prompt:

```
Please generate a valid relation path that can be helpful for
answering the following question:
{Question}
```

For LLM prompting during reasoning (Section 4.3), we use the following prompt:

```
Based on the reasoning paths, please answer the given question.
Please keep the answer as simple as possible and return all the
possible answers as a list.\n
Reasoning Paths:  {Reasoning Paths} \n
Question:  {Question}
```

During GNN inference, each node in the subgraph is assigned a probability of being the correct answer, which is normalized via $\mathrm{softmax}$. To retrieve answer candidates, we sort the nodes based on the their probability scores, and select the top nodes whose cumulative probability score is below a threshold. We set the threshold to 0.95. To retrieve the shortest paths between the question entities and answer candidates for RAG, we use the NetworkX library[4].

**Competing Approaches**.

---

[1]https://github.com/cmavro/ReaRev_KGQA

[2]https://github.com/RManLuo/reasoning-on-graphs

[3]https://github.com/RUCKBReasoning/SubgraphRetrievalKBQA

[4]https://networkx.org/

We evaluate the following categories of methods: 1. Embedding, 2. GNN, 3. LLM, 4. KG+LMM, and 5. GNN+LLM.

1. KV-Mem Miller et al. (2016) is a key-value memory network for KGQA. EmbedKGQA Saxena et al. (2020) utilizes KG pre-trained embeddings Trouillon et al. (2016) to improve multi-hop reasoning. TransferNet Shi et al. (2021) improves multi-hop reasoning over the relation set. Rigel Sen et al. (2021) improves reasoning with questions of multiple entities.

2. GraftNet Sun et al. (2018) uses a convolution-based GNN Kipf & Welling (2016). PullNet Sun et al. (2019) is built on top of GraftNet, but learns which nodes to retrieve via selecting shortest paths to the answers. NSM He et al. (2021) is the adaptation of GNNs for KGQA. NSM+h He et al. (2021) improves NSM for multi-hop reasoning. SQALER Atzeni et al. (2021) learns which relations (facts) to retrieve during KGQA for GNN reasoning. Similarly, SR+NSM (Zhang et al., 2022a) proposes a relation-path retrieval. UniKGQA (Jiang et al., 2023b) unifies the graph retrieval and reasoning process with a single LM. ReaRev (Mavromatis & Karypis, 2022) explores diverse reasoning paths in a multi-stage manner.

3. We experiment with instruction-tuned LLMs. Flan-T5 (Chung et al., 2024) is based on T5, while Aplaca (Taori et al., 2023) and LLaMA2-Chat (Touvron et al., 2023) are based on LLaMA. ChatGPT[5] is a powerful closed-source LLM that excels in many complex tasks. ChatGPT+CoT uses the chain-of-thought (Wei et al., 2022) prompt to improve the ChatGPT. We access ChatGPT `gpt-3.5-turbo` through its API (as of May 2024).

4. KD-CoT (Wang et al., 2023) enhances CoT prompting for LLMs with relevant knowledge from KGs. StructGPT (Jiang et al., 2023a) retrieves KG facts for RAG. KB-BINDER (Li et al., 2023) enhances LLM reasoning by generating logical forms of the questions. ToG (Sun et al., 2024) uses a powerful LLM to select relevant facts hop-by-hop. RoG (Luo et al., 2024) uses the LLM to generate relation paths for better planning.

5. G-Retriever (He et al., 2024) augments LLMs with GNN-based prompt tuning.

**Evaluation metric discussion**. We clarify the evaluation metrics in Table 2. H@1 evaluation assumes that we are given a list of scored candidate answers (sorted based on the model's scores). However, since LLMs generate free-form answers, their responses can include multiple answers, which complicates the direct application of Hit@1. For example, consider the following hypothesized case:

> Question: What do Jamaican people speak?
> Answer: English
> LLM Response: Jamaican people speak French and English.

In this case, the Hit score would be 1.0, as "English" is included in the response, although the LLM generates the incorrect response "French". This is the score that prior methods report as Hit@1 for LLMs. However, if we were to treat the LLM response as a list [French, English], the Hit@1 score would be 0.0, because the answer at rank 1 (French) is not the correct one.

For this reason, we do not combine H@1 and Hit metrics for LLMs, as doing so could lead to an *artificially inflated performance*, and report LLM performance separately based on the Hit metric.

## E  ADDITIONAL EXPERIMENTAL RESULTS

### E.1  QUESTION ANALYSIS

Following the case studies presented in Figure 5 and Figure 6, we provide numerical results on how GNN-RAG improves multi-hop question answering and how retrieval augmentation (RA) enhances simple hop questions. Table 9 summarizes these results. GNN-RAG improves performance on multi-hop questions (≥2 hops) by 6.5–11.8% F1 points over RoG. Furthermore, RA improves performance on single-hop questions by 0.8–2.6% F1 points over GNN-RAG.

Table 10 presents results with respect to the number of correct answers. As shown, RA enhances GNN-RAG in almost all cases as it can fetch correct answers that might have been missed by the GNN.

---

[5]https://openai.com/blog/chatgpt

Table 9: Performance analysis (F1) based on the number of maximum hops that connect question entities to answer entities.

| Method | WebQSP | | | CWQ | | |
|---|---|---|---|---|---|---|
| | 1 hop | 2 hop | ≥3 hop | 1 hop | 2 hop | ≥3 hop |
| RoG | 73.4 | 63.3 | – | 50.4 | 60.7 | 40.0 |
| GNN-RAG | 72.0 | 69.8 | – | 47.4 | 69.4 | 51.8 |
| GNN-RAG +RA | 74.6 | 71.1 | – | 48.2 | 70.9 | 47.7 |

Table 10: Performance analysis (F1) based on the number of answers (#Ans).

| Method | WebQSP | | | | CWQ | | | |
|---|---|---|---|---|---|---|---|---|
| | #Ans=1 | 2≤#Ans≤4 | 5≤#Ans≤9 | #Ans≥10 | #Ans=1 | 2≤#Ans≤4 | 5≤#Ans≤9 | #Ans≥10 |
| RoG | 67.89 | 79.39 | 75.04 | 58.33 | 56.90 | 53.73 | 58.36 | 43.62 |
| GNN-RAG | 71.24 | 76.30 | 74.06 | 56.28 | 60.40 | 55.52 | 61.49 | 50.08 |
| GNN-RAG +RA | 71.16 | 82.31 | 77.78 | 57.71 | 62.09 | 56.47 | 62.87 | 50.33 |

## E.2 GNN EFFECT

Table 11: Performance comparison of different GNN models at KGQA (extended).

| Retriever | KGQA Model | WebQSP | | | CWQ | | |
|---|---|---|---|---|---|---|---|
| | | Hit* | H@1 | F1 | Hit* | H@1 | F1 |
| Dense Subgraph | GraftNet | – | 66.7 | 62.4 | – | 45.3 | 35.8 |
| Dense Subgraph | NSM | – | 68.7 | 62.8 | – | 47.9 | 42.0 |
| Dense Subgraph | ReaRev | – | 76.4 | 70.9 | – | 52.7 | 49.1 |
| Dense Subgraph | LLaMA2-Chat-7B (tuned) | – | 68.7 | 54.3 | – | 45.5 | 41.9 |
| RoG | LLaMA2-Chat-7B (tuned) | 85.7 | 80.0 | 70.8 | 62.6 | 57.8 | 56.2 |
| GNN-RAG: GraftNet | | 82.8 | 78.6 | 69.8 | 58.2 | 51.9 | 49.4 |
| GNN-RAG: NSM | | 85.0 | 79.6 | 70.4 | 58.5 | 52.5 | 50.1 |
| GNN-RAG: ReaRev | | 85.7 | 80.6 | 71.3 | 66.8 | 61.7 | 59.4 |

GNN-RAG employs ReaRev (Mavromatis & Karypis, 2022) as its GNN retriever, which is a powerful GNN for deep KG reasoning. In this section, we ablate on the impact of the GNN used for retrieval, i.e., how strong and weak GNNs affect KGQA performance. We experiment with GraftNet (Sun et al., 2018) and NSM (He et al., 2021) GNNs, which are less powerful than ReaRev at KGQA. The results are presented in Table 11. As shown, strong GNNs (ReaRev) are required in order to improve RAG at KGQA. Retrieval with weak GNNs (NSM and GraftNet) underperfoms retrieval with ReaRev by up to 9.8% and retrieval with RoG by up to 5.9% points at H@1.

## E.3 RETRIEVAL AUGMENTATION

Table 12 has the extended results of Table 4, showing performance results on all three metrics (Hit / H@1 / F1) with respect to the retrieval method used. Overall, GNN-RAG improves the vanilla LLM by 149–182%, when employing the same number of LLM calls for retrieval.

## E.4 PROMPT ABLATION

When using RAG, LLM performance depends on the prompts used. To ablate on the prompt impact, we experiment with the following prompts:

- **Prompt A**:

```
Based on the reasoning paths, please answer the
given question.  Please keep the answer as
simple as possible and return all the possible
answers as a list.\n
Reasoning Paths:  {Reasoning Paths} \n
Question:  {Question}
```

Table 12: Performance comparison of retrieval augmentation approaches (extended).

| Retriever | KGQA Model | #LLM Calls (total) | WebQSP Hit* | WebQSP H@1 | WebQSP F1 | CWQ Hit* | CWQ H@1 | CWQ F1 | Avg. |
|---|---|---|---|---|---|---|---|---|---|
| Dense Subgraph | (i) ReaRev + SBERT | 0 | – | 76.4 | 70.9 | – | 52.9 | 47.8 | – |
|  | (ii) ReaRev + LM$_{SR}$ | 0 | – | 77.5 | 72.8 | – | 52.7 | 49.1 | – |
| None | | 1 | 65.6 | 60.4 | 49.7 | 40.1 | 36.2 | 33.8 | 47.63 |
| (iii) LLM-based | LLaMA2-Chat-7B (tuned) | 4 | 85.7 | 80.0 | 70.8 | 62.6 | 57.8 | 56.2 | 68.85 |
| GNN-RAG: (i) | | 1 | 85.7 | 80.6 | 71.3 | 66.8 | 61.7 | 59.4 | 70.92 |
| GNN-RAG: (ii) | | 1 | 85.0 | 80.3 | 71.5 | 66.2 | 61.3 | 58.9 | 70.50 |
| GNN-RAG: (i) + (ii) | | 1 | 87.2 | 81.0 | 71.7 | 65.5 | 59.5 | 57.5 | 70.40 |
| GNN-RAG: (i) + (iii) | LLaMA2-Chat-7B (tuned) | 4 | **90.7** | **82.8** | **73.5** | **68.7** | 62.8 | 60.4 | **73.15** |
| GNN-RAG: (ii) + (iii) | | 4 | 89.9 | 82.4 | 73.4 | 67.9 | **63.0** | **61.0** | 72.93 |
| GNN-RAG: (i) + (ii) + (iii) | | 4 | 90.1 | 81.7 | 72.3 | 67.3 | 61.5 | 59.1 | 72.00 |
| None | | 1 | 64.4 | – | – | 34.6 | – | – | – |
| GNN-RAG: (i) + (ii) | LLaMA2-Chat-7B | 1 | 86.8 | – | – | 62.9 | – | – | – |
| GNN-RAG: (i) + (iii) | | 4 | 88.5 | – | – | 62.1 | – | – | – |

- **Prompt B**:

```
Based on the provided knowledge, please answer
the given question.  Please keep the answer as
simple as possible and return all the possible
answers as a list.\n
Knowledge:  {Reasoning Paths} \n
Question:  {Question}
```

- **Prompt C**:

```
Your tasks is to use the following facts
and answer the question.
Make sure that you use the information
from the facts provided. Please keep the answer
as simple as possible and return all the
possible answers as a list.\n
The facts are the following:  {Reasoning Paths}
\n
Question:  {Question}
```

Table 13: Performance comparison (%Hit) based on different input prompts.

|  |  | WebQSP | CWQ |
|---|---|---|---|
| Prompt A | RoG | 84.8 | 56.4 |
|  | GNN-RAG | 86.8 | 62.9 |
| Prompt B | RoG | 84.3 | 55.2 |
|  | GNN-RAG | 85.2 | 61.7 |
| Prompt C | RoG | 81.6 | 51.8 |
|  | GNN-RAG | 84.4 | 59.4 |

We provide the results based on different input prompts in Table 13. As the results indicate, GNN-RAG outperforms RoG in all cases, being robust at the prompt selection.

### E.5 EFFECT OF TRAINING DATA

**Training Cost**. GNN-RAG requires only fine-tuning the GNN for retrieval. The downstream LLM can be fine-tuned (our default implementation) or not (as we experimented with in Table 6). Fine-tuning the downstream LLM is memory-intensive. For example, if we use 2 A100-80G GPUs, 1 epoch of

30k training data requires more than 12 hours. GNN training is much more efficient: On a GeForce RTX 3090, 1 epoch of 30k training data needs less than 15 minutes and less than 8GB of GPU memory.

Table 14: Impact of LLM tuning.

| Retrieval | LLM | WebQSP | CWQ |
|-----------|-----|--------|-----|
| RoG | LLaMa2-Chat-7B (untuned) | 84.8 | 56.4 |
| RoG | LLaMa2-Chat-7B (tuned) | 85.7 | 62.6 |
| GNN-RAG | LLaMa2-Chat-7B (untuned) | 85.2 | 62.7 |
| GNN-RAG | LLaMa2-Chat-7B (tuned) | 85.7 | 66.7 |

**Data Size Impact**. Fine-tuning the downstream LLM generally improves performance. In Table 14, we compare LLaMa2-Chat-7B and LLaMa2-Chat-7B fine-tuned. As shown (Hit metric), GNN-RAG demonstrates a more stable performance when switching between the two LLMs. Specifically, GNN-RAG experiences a relatively small drop of 0.5-5.0 points, whereas RoG suffers from a larger performance degradation of 0.9-6.2 points under the same conditions. CWQ has more data (27.6k) than WebQSP (2.8k) and thus, performance improvement when using the tuned LLM is larger.

Table 15: Number of training data impact on CWQ.

| Retrieval | # Training Data | CWQ Hit (%) |
|-----------|-----------------|-------------|
| RoG | 30k | 62.6 |
| GNN-RAG | 27.6k | 66.7 |
| GNN-RAG | 10k | 63.7 |

In Table 15, we provide results when we use 10k training data of CWQ when training the GNN. As shown, although GNN-RAG uses approximately 3x less data, it still outperforms RoG (which uses 30k data from both CWQ and WebQSP for training).

Table 16: Performance results based on different training data.

| Method | WebQSP | | | CWQ | | |
|--------|--------|--|--|-----|--|--|
| | Training Data (Retriever) | Training Data (KGQA Model) | Hit | Training Data (Retriever) | Training Data (KGQA Model) | Hit |
| UniKGQA | WebQSP | WebQSP | 77.2 | CWQ | CWQ | 51.2 |
| RoG | WebQSP | WebQSP | 81.5 | CWQ | CWQ | 59.1 |
| | WebQSP+CWQ | None | 84.8 | WebQSP+CWQ | None | 56.4 |
| | WebQSP+CWQ | WebQSP+CWQ | 85.7 | WebQSP+CWQ | WebQSP+CWQ | 62.6 |
| GNN-RAG | WebQSP | None | 86.8 | CWQ | None | 62.9 |
| | WebQSP | WebQSP+CWQ | **87.2** | CWQ | WebQSP+CWQ | **66.8** |

Table 16 compares performance of different methods based on the training data used for training the retriever and the KGQA model. For example, GNN-RAG trains a GNN model for retrieval and uses a LLM for KGQA, which can be fine-tuned or not. As the results show, GNN-RAG outperforms the competing methods (RoG and UniKGQA) by either fine-tuning the KGQA model or not, while it uses the same or less data for training its retriever.

## E.6 GRAPH EFFECT

GNNs operate on dense subgraphs, which might include noisy information. A question that arises is whether removing irrelevant information from the subgraph would improve GNN retrieval. We experiment with SR (Zhang et al., 2022a), which learns to prune question-irrelevant facts from the KG. As shown in Table 17, although SR can improve the GNN reasoning results – see row (a) vs. (b) at CWQ –, the retrieval effectiveness deteriorates; rows (c) and (d). After examination, we found that the sparse subgraph may contain disconnected KG parts. In this case, GNN-RAG's extraction of the shortest paths fails, and GNN-RAG returns empty KG information.

Table 17: Performance comparison on different subgraphs.

| Retriever | KGQA Model | WebQSP | | | CWQ | | |
|---|---|---|---|---|---|---|---|
| | | Hit* | H@1 | F1 | Hit* | H@1 | F1 |
| a) Dense Subgraph | (A) ReaRev + LM$_{SR}$ | – | 77.5 | 72.8 | – | 52.7 | 49.1 |
| b) Sparse Subgraph (Zhang et al., 2022a) | (B) ReaRev + LM$_{SR}$ | – | 74.2 | 69.8 | – | 53.3 | 49.7 |
| c) GNN-RAG: (A) | LLaMA2-Chat-7B (tuned) | 85.0 | 80.3 | 71.5 | 66.2 | 61.3 | 58.9 |
| d) GNN-RAG: (B) | | 83.4 | 78.9 | 69.8 | 60.6 | 55.6 | 53.3 |

### E.7 FURTHER ABLATIONS

Regarding GNN hyperparameters, we provide sensitivity analysis on the number of GNN layers $L$ in Table 1, which shows that deep GNNs are better retrievers for mutli-hop KGQA.

As an additional ablation study, we set the threshold $\theta$, which controls the number of candidate answer nodes for entity selection, to 0.99 (retrieves more candidate answers), to 0.95 (default), and to 0.75 (retrieves less candidate answers). GNN-RAG performance is shown in Table 18. Increasing the threshold (0.99) to retrieve more context, can further increase performance to 85.9%. Lower threshold (0.75) might miss some answers and the performance drops to 83.5%.

Table 18: Threshold $\theta$ impact for answer node selection (WebQSP Hit %).

| | $\theta = 0.99$ | $\theta = 0.95$ | $\theta = 0.75$ |
|---|---|---|---|
| GNN-RAG | 85.9 | 85.7 | 83.8 |

## F  LIMITATIONS

GNN-RAG assumes that the KG subgraph, on which the GNN reasons, contains answer nodes. However, this may not be true for all questions or when errors in entity linking happen. In addition, GNN-RAG employs simple prompting with the shortest paths from question entities to candidate answers as context. As an extension, GNN-RAG can be combined with prompt optimization (Wen et al., 2023; Zhang et al., 2023a) so that the LLM understands the graph better. Moreover, similar to conventional retrieval which focuses on identifying relevant information (text documents or KG nodes in Figure 1) regardless the downstream LLM, the scope of our GNN-RAG contributions is to improve the retrieval results over the KG without specialized GNN-LLM interactions. However, the GNN and the LLM could be coupled via iterative retrieval (Asai et al., 2023) to further improve KGQA.

## G  BROADER IMPACTS

GNN-RAG is a method that grounds the LLM generations for QA using ground-truth facts from the KG. As a result, GNN-RAG can have positive societal impacts by using KG information to alleviate LLM hallucinations in tasks such as QA.

