# OpenReview forum: "GNN-RAG: Graph Neural Retrieval for Large Language Model Reasoning"
_ICLR.cc/2025/Conference — Submitted to ICLR 2025_

### Official Review · Reviewer_TceY · 2024-10-21

**Soundness:** 4
**Presentation:** 3
**Contribution:** 2
**Rating:** 6
**Confidence:** 5

**Summary:**

This paper proposes a Retrieve-augmented generation (RAG) framework named GNN-RAG for the knowledge graph question answering~(KGQA) task. GNN-RAG first employs a GNN to find plausible candidate answers for the given question, and extract the shortest path between the topic entity and candidate answers as reasoning paths. Then, the framework finetunes and employs an LLM to answer the question based on these candidate entities and related reasoning paths. GNN-RAG achieves the SOTA performance on the WebQSP and the CWQ datasets. The paper also makes direct and detailed comparisons between the proposed method and its competitive baseline RoG.

**Strengths:**

1)  Key Original Contribution: This paper proposes to utilize GNN to retrieve and select proper candidate answers for the given question, which effectively alleviates the limitation of LLM-based KGQA methods. Note: The Key contribution belongs to the “Retrieval” part.

2) Compared to LLM-based baseline methods, GNN-RAG is simple and effective. It achieves state-of-the-art performance on WebQSP with Llama2-7B model. The performance on CWQ is slightly lower than ToG with GPT4 as the backbone LLM.

3) The authors of this paper conducted detailed comparisons between RoG and the proposed GNN-RAG, and examined the effectiveness of GNN-RAG, RoG, and ToG on different kinds of LLMs.

**Weaknesses:**

1) Although the proposed method demonstrates satisfactory performance, it is “relatively” incremental compared to ToG and RoG. The retrieval of reasoning paths with GNNs is not a brand-new idea in the QA domain [1]. Apart from the method of path retrieval, less essential differences can be found between RoG and GNN-RAG. Nevertheless, this reviewer does not believe that the paper should be rejected based solely on this reason.

2) This paper lacks a detailed explanation of the design of the GNN. Specifically, it could be beneficial to explicitly model the classification step with equation(s). This reviewer does not know how the authors perform the classification with the entity embeddings outputted by the L-th layer. He can only make an educated guess that the classification operation is conducted by an FC layer followed by a softmax / sigmoid layer.

3) It could be beneficial to take UniKGQA (or ReaRev) and/or ToG into comparison in Table 3 and 4 since Figure 3 mentioned the landscape of three types of existing methods.

4) The paper did not discuss the time cost for LLM fine-tuning and inference. This reviewer is surprised to see that the proposed method, with a 7B model, outperforms baselines with a 70B model or GPT. However, the efficiency of the proposed method cannot be judged solely by the number of LLM calls required during the inference stage. GPU hours for LLM fine-tuning is also an important metric.

[1] Relation-aware Bidirectional Path Reasoning for Commonsense Question Answering (CoNLL
2021)

**Questions:**

1) How does GNN-RAG initialize the entity / relation embeddings? It would be beneficial to explicitly state whether the framework uses randomly initialized or pre-trained embeddings, or whether these embeddings are optimized with GNN parameters. '
*(This is crucially important especially for the IC"LR" conference.)*

2) How to emphasize k in equation 3? Is that means, given a fixed LM, we can have k different embeddings for a specific question? Or, the question embedding is pooled based on the final layer output of each of the tokens? Please *explicitly* state the structure of the mentioned "attention-based pooling neural network" with equations.

This reviewer sincerely requests the authors to enrich the implementation details in section 4.1. The current submission is insufficient for readers to reproduce the experimental results without checking the source code. Spaces are available to add these details since the paper does not fill 10 pages.

3) Would you mind adding a small section and/or a table to discuss the time cost of LLM fine-tuning?

---

### Official Review · Reviewer_7Tku · 2024-10-22

**Soundness:** 3
**Presentation:** 4
**Contribution:** 2
**Rating:** 6
**Confidence:** 4

**Summary:**

The paper introduces GNN-RAG, a KGQA (Knowledge Graph Question Answering) algorithm that combines GNN (Graph Neural Networks) with LLM reasoning. This method uses a GNN-based KGQA algorithm to retrieve reasoning evidence from a KG, allowing the LLM to generate the final answer. Experiments show that this approach effectively improves the performance of GNN-based KGQA methods and outperforms other LLM-based KGQA algorithms in terms of accuracy and efficiency.

**Strengths:**

1. GNN-RAG uses a GNN-based KGQA algorithm to obtain candidate answers to the question, then samples the shortest path between the topic entity and candidate entities as reasoning evidence, allowing the LLM to generate the final answer. Experimental results demonstrate that the GNN-based KGQA algorithm can effectively retrieve reasoning evidence from the KG, and the fine-tuned LLM can accurately infer results from the evidence.

2. Compared to existing LLM-based KGQA methods, GNN-RAG shows better efficiency and accuracy on two KGQA datasets.

3. The paper provides thorough case studies and theoretical analysis on the strengths and weaknesses of the GNN approach.

4. The structure of the paper is clear and easy to follow.

**Weaknesses:**

1. Using existing KGQA methods to retrieve information from a KG and then allowing a LLM to perform reasoning based on the retrieved information is a rather intuitive idea. In some LLM-based KG completion works, similar approaches have been used, where traditional KG completion models retrieve relevant triples and candidate entities from the KG, and then LLMs use their semantic understanding to perform reasoning [1][2]. This work seems to have simply applied a similar idea to the KGQA task. The retrieval stage follows existing GNN-based KGQA methods without providing sufficient theoretical analysis of the GNN's advantages in multi-hop information retrieval, and the utility of GNN-based KGQA methods is also intuitive since GNN models currently perform the best among existing IR-based KGQA methods. Moreover, the reasoning stage follows the design of RoG [3], so the innovation of this method is rather limited.

2. This approach requires training two models, namely a KBQA model and fine-tuning a LLM. While the approach has an efficiency advantage during inference, it demands a large amount of labeled training data, which is difficult to obtain in real-world scenarios. The paper does not sufficiently discuss the costs of model training, such as how much data is used to fine-tune the LLM, the time required to train the KBQA model, or how changes in training data affect the method's performance.

3. As shown in Table 5, this method is constrained by the GNN-based KGQA model, and when the performance of the GNN-based KGQA model is poor, the overall model's performance declines significantly. This point is also mentioned in the limitations section of the paper.

[1] Wei, Yanbin, et al. "KICGPT: Large Language Model with Knowledge in Context for Knowledge Graph Completion." Findings of the Association for Computational Linguistics: EMNLP 2023. 2023.
[2] Liu, Yang, et al. "Finetuning Generative Large Language Models with Discrimination Instructions for Knowledge Graph Completion." arXiv preprint arXiv:2407.16127 (2024).
[3] LUO, LINHAO, et al. "Reasoning on Graphs: Faithful and Interpretable Large Language Model Reasoning." The Twelfth International Conference on Learning Representations.

**Questions:**

1. What is the cost of model training? For example, how much data was used to fine-tune the large language model, and how long did it take to train the KBQA model?

2. How does changing the amount of training data (either fine-tuning data or KBQA model training data) affect the performance of the method?

3. In Table 6, are the results of Alpaca-7b based on the Alpaca-7b model fine-tuned for this task, or are they based on the original Alpaca-7b model? Would fine-tuning a more powerful LLM yield better results?

4. If only RA is used for knowledge retrieval, how would the performance change?

5. For the KGQA task, constructing QA training data versus constructing QA data with SPARQL queries is not significantly different since the groud truth answers need to be validated with SPARQL queries. If SOTA SP-based KGQA methods are used for obtaining candidate entities, would its performance be superior to that of the GNN-based method, especially in cases of 1-hop queries?

---

### Official Review · Reviewer_EGFX · 2024-10-30

**Soundness:** 3
**Presentation:** 3
**Contribution:** 3
**Rating:** 8
**Confidence:** 3

**Summary:**

This paper present GNN-RGA, a novel graph neural method for enhancing RAG in KGQA with GNNs. GNN-RAG achieves state-of-the-art performance in two widely used KGQA benchmarks (WebQSP and CWQ).

**Strengths:**

1. This is a solid study with reasonable technical contributions. The large number of baseline models used in the experiments is impressive.
2. The paper is generally understandable and clearly explains the technical parts to a certain extent.
3. The figures and charts in the manuscript are exceptionally clear and well-presented.

**Weaknesses:**

1. This paper does not provide sufficient details on the RAG WITH LLM.
2. The paper misses many related studies such as [1][2][3][4], which could provide a broader context and highlight the novelty and contribution of GNN-RGA more effectively.
[1] ARL: An adaptive reinforcement learning framework for complex question answering over knowledge base. Inf. Process. Manag. 59(3): 102933 (2022)
[2] Query Path Generation via Bidirectional Reasoning for Multihop Question Answering From Knowledge Bases. IEEE Trans. Cogn. Dev. Syst. 15(3): 1183-1195 (2023)
[3] Question-Directed Reasoning With Relation-Aware Graph Attention Network for Complex Question Answering Over Knowledge Graph. IEEE ACM Trans. Audio Speech Lang. Process. 32: 1915-1927 (2024)

**Questions:**

1. This paper does not provide sufficient details on the RAG WITH LLM.
2. The paper misses many related studies such as [1][2][3][4], which could provide a broader context and highlight the novelty and contribution of GNN-RGA more effectively.
[1] ARL: An adaptive reinforcement learning framework for complex question answering over knowledge base. Inf. Process. Manag. 59(3): 102933 (2022)
[2] Query Path Generation via Bidirectional Reasoning for Multihop Question Answering From Knowledge Bases. IEEE Trans. Cogn. Dev. Syst. 15(3): 1183-1195 (2023)
[3] Question-Directed Reasoning With Relation-Aware Graph Attention Network for Complex Question Answering Over Knowledge Graph. IEEE ACM Trans. Audio Speech Lang. Process. 32: 1915-1927 (2024)

---

### Official Review · Reviewer_3FK4 · 2024-10-30

**Soundness:** 2
**Presentation:** 3
**Contribution:** 2
**Rating:** 3
**Confidence:** 4

**Summary:**

The paper proposes a GNN-RAG framework that combines GNN and LLM for KGQA task. GNN-RAG first leverages SOTA GNN model to retrieve answer candidates for a given question, then the shortest paths that connect question entities and answer candidates are extracted to represent KG reasoning paths for LLM’s final output. Experiments demonstrate that the GNN-RAG with fine-tuned LLaMA2-7B can achieve state-of-the-art performance.

**Strengths:**

1. The paper propose to leverage GNN as a retriever to improve RAG in complex KGQA.
2. GNN-RAG improves LLMs without incurring additional LLM calls, and achieves competitive performance.
3. The authors provide codes for their implementation.

**Weaknesses:**

1. The overall framework lacks novelty and is just like a simple concatenation of existing methods.

2. The role of GNN as a retriever is only to identify candidate entities, rather than to "learn useful graph information" as stated by the author.

3. There are a large number of blank spaces in the experimental result table.

**Questions:**

1. In GNN-RAG framework, the role of GNN is merely to obtain high-scoring candidate entities. Can the GNN be replaced by any other model, such as a simple but effective embedding-based model? What is the necessity of having a GNN in this framework?

2. When there is more than one shortest path between the candidate entity and the question entity, how should the retrieval result be determined? How can you ensure that the shortest path accurately aligns with the semantic information of the question?

3. Is the setting of 'RA' in the method equivalent to performing an ensemble between GNN-RAG and RoG?

4. Many LLM-based methods typically report their results using the Hits@1 metric, but you place their results under the Hit metric. By doing so, larger performance improvements are observed. Is there sufficient evidence to suggest that this comparison is appropriate?

---

### Official Review · Reviewer_sWgu · 2024-11-01

**Soundness:** 2
**Presentation:** 3
**Contribution:** 2
**Rating:** 5
**Confidence:** 5

**Summary:**

The paper presents GNN-RAG, a framework aimed at improving retrieval-augmented generation for KGQA. GNN-RAG leverages Graph Neural Networks for effective graph-based reasoning and retrieval, supplying candidate reasoning paths as context to the language model. Experimental results show that GNN-RAG, with a tuned 7B LLM, outperforms GPT-4 in performance.

**Strengths:**

1. The paper is clearly presented and easy to follow.
2. The GNN-RAG approach integrates seamlessly with various LLMs.
3. The proposed method shows notable improvements in the KBQA task.

**Weaknesses:**

1. The contribution of the paper feels limited; the approach mainly leverages LLMs to retrieve the correct answers from REAREV, without significant novelty in methodology.
2. Experiments are conducted on only three datasets, with numerous competitor results missing, which limits comprehensive evaluation.

**Questions:**

1. Why are many results missing in Table 2? In addition, expanding the experiments to include more datasets would help demonstrate the generalizability of the proposed method.
2. The model involves several hyperparameters (e.g., entity selection threshold and m), yet sensitivity analysis is absent. Including such analysis would improve the robustness of the evaluation.
3. According to the original REAREV method, H@1 on the MetaQA-3 dataset achieves 98.9, whereas performance with the LLM (i.e., GNN+RAG) in Table 3 decreases to 98.6. Could the authors clarify in which cases the integration with the LLM might lead to a performance drop?

---

### Meta-Review · Area_Chair_1sx4 · 2024-12-24

**Metareview:**

Summary: The paper proposes GNN-RAG, a framework combining Graph Neural Networks (GNNs) and Retrieval-Augmented Generation (RAG) for Knowledge Graph Question Answering (KGQA). The framework uses GNNs for retrieval of graph-based reasoning paths, which are then provided as context to downstream Large Language Models (LLMs). Experimental studies are conducted on two KGQA benchmarks (WebQSP and CWQ), outperforming or matching GPT-4 performance using a smaller LLM (7B parameters) and retrieval.

Strengths:
- Clear presentation with thorough explanations and illustrative examples
- Detailed ablation studies and analysis of different components

Weakness:
- Lukewarm response from all but one reviewer and the positive reviewer didn't champion the paper
- Limited Novelty: Several reviewers noted that the approach primarily combines existing GNN and RAG techniques, raising concerns about methodological originality.
- Role of GNNs: The GNN primarily retrieves high-scoring candidate entities rather than performing more complex graph reasoning, which some reviewers found underwhelming.
- Missing important baseline: Apart from the works pointed out by reviewer, there are several missing important works, e.g. Das et al "Case-Based Reasoning for Natural Language Queries over Knowledge Bases", 2021. He et al. "Improving Multi-hop Knowledge Base Question Answering by Learning Intermediate Supervision Signals", 2022, and inter alia.
- Incomplete empirical comparisons and stronger than substantiated claims: In light of other missing works, it is not clear if GNN-RAG achieves SoTA results

Decision: Given the lack of enthusiasm from the reviewers, incremental novelty, and missing prior works, unfortunately, the paper can't be accepted in its current form and addressing all the concerns would warrant another round of reviewing.

**Additional Comments On Reviewer Discussion:**

We thank the authors for engaging during the discussion phase towards improving the paper. Below are some of the highlights:

1. Novelty concerns: Multiple reviewers questioned the technical innovation. The authors argued that showing GNNs remain effective complementary components to LLMs is an important contribution, though some reviewers remained unconvinced.
2. Missing implementation details: Reviewers requested more details about GNN architecture, embeddings, etc. The authors provided comprehensive additional technical details in their response and updated appendix.
3. Baseline comparisons: Reviewers noted some missing baseline comparisons. The authors clarified that some baselines (e.g., ToG) had reproducibility issues, but agreed to add additional comparisons where possible.
4. Training costs/efficiency: Reviewers requested more discussion of computational requirements. The authors added details about training times and resource usage.

The authors were responsive and provided detailed clarifications and additional results. However, some of the main concerns, e.g. novelty, weren't fully resolved.

---

### Decision · Program_Chairs · 2025-01-22

Reject